# A Human Organoid Model of Aggressive Hepatoblastoma for Disease Modeling and Drug Testing

**DOI:** 10.3390/cancers12092668

**Published:** 2020-09-18

**Authors:** James A. Saltsman, William J. Hammond, Nicole J. C. Narayan, David Requena, Helmuth Gehart, Gadi Lalazar, Michael P. LaQuaglia, Hans Clevers, Sanford Simon

**Affiliations:** 1Laboratory of Cellular Biophysics, The Rockefeller University, 1230 York Avenue, New York, NY 10065, USA; james.saltsman@umontreal.ca (J.A.S.); william.hammond3@ucalgary.ca (W.J.H.); njc9006@nyp.org (N.J.C.N.); drequenaan@rockefeller.edu (D.R.); glalazar@rockefeller.edu (G.L.); 2Pediatric Surgery Service, Department of Surgery, Memorial Sloan-Kettering Cancer Center, 1275 York Avenue, New York, NY 10065, USA; laquaglm@mskcc.org; 3Hubrecht Institute, KNAW and University Medical Center Utrecht, 3584CT Utrecht, The Netherlands; h.gehart@hubrecht.eu (H.G.); h.clevers@hubrecht.eu (H.C.); 4The Princess Maxima Center for Pediatric Oncology, 3584CT Utrecht, The Netherlands

**Keywords:** pediatrics, oncology, pediatric oncology, pediatric solid tumor, liver cancer, hepatoblastoma, 3-D culture, organoids, sequencing

## Abstract

**Simple Summary:**

Hepatoblastoma is the most common childhood liver cancer, making up over 90% of malignant liver tumors in children younger than 5 years of age. Currently, research to find new treatments for treatment-resistant hepatoblastoma is limited by a lack of appropriate models to study the disease. In this study, we describe a novel patient-derived organoid model of aggressive hepatoblastoma that can be used to study the disease in the laboratory and test new treatments. We demonstrate that tumor organoids share the same genomic profile as the patient tumors from which they are derived, and also demonstrate similar features with respect to gene expression profiles and beta-catenin signaling. We also demonstrate the feasibility of using hepatoblastoma organoids to complete a drug screen alongside normal liver control organoids derived from the same patient, and report promising initial results of anti-tumor activity of the BET inhibitor JQ1.

**Abstract:**

Hepatoblastoma is the most common childhood liver cancer. Although survival has improved significantly over the past few decades, there remains a group of children with aggressive disease who do not respond to current treatment regimens. There is a critical need for novel models to study aggressive hepatoblastoma as research to find new treatments is hampered by the small number of laboratory models of the disease. Organoids have emerged as robust models for many diseases, including cancer. We have generated and characterized a novel organoid model of aggressive hepatoblastoma directly from freshly resected patient tumors as a proof of concept for this approach. Hepatoblastoma tumor organoids recapitulate the key elements of patient tumors, including tumor architecture, mutational profile, gene expression patterns, and features of Wnt/β-catenin signaling that are hallmarks of hepatoblastoma pathophysiology. Tumor organoids were successfully used alongside non-tumor liver organoids from the same patient to perform a drug screen using twelve candidate compounds. One drug, JQ1, demonstrated increased destruction of liver organoids from hepatoblastoma tumor tissue relative to organoids from the adjacent non-tumor liver. Our findings suggest that hepatoblastoma organoids could be used for a variety of applications and have the potential to improve treatment options for the subset of hepatoblastoma patients who do not respond to existing treatments.

## 1. Introduction

Hepatoblastoma is the most common childhood liver cancer, making up 80% of all malignant liver neoplasms in children, with over 90% of such lesions being in children younger than 5 years of age [1]. The development of hepatoblastoma has been linked to environmental factors such as prematurity and genetic syndromes including Beckwith–Wiedemann syndrome [2]. Recent evidence suggests that the incidence rate of hepatoblastoma is increasing by more than 4% annually [3]. Unlike hepatocellular carcinoma—the most common liver malignancy in adults—hepatoblastoma is an embryonal neoplasm believed to arise from hepatic precursor cells (hepatoblasts). Tumors display heterogeneous histologic and molecular features that appear similar to patterns present at various stages of liver development [4]. Aberrations of Wnt/β-catenin signaling are key to the pathogenesis of the disease [2,4,5,6]. Although survival for hepatoblastoma has improved significantly over the past few decades, there remains a group of children who experience poor outcomes, including death. These patients are not eligible for surgical resection and do not respond to current chemotherapy regimens. In order to develop improved therapies for such children, additional research is required to understand the biology of aggressive hepatoblastoma and perform targeted drug screening. Currently, research of this nature is limited by the small number of in vitro models that accurately reflect the disease [7,8]. In particular, there is a specific dearth of models that reflect the aggressive subset of tumors that lead to the majority of deaths from hepatoblastoma [7]. 

Organoids have emerged as model systems for a variety of tissue types and disease states. Organoids are generally defined as three-dimensional cell culture systems that recapitulate tissue architecture and function. They can be derived from induced pluripotent stem cells (iPSCs) or from adult stem cells (ASCs) present in mature tissues [9]. Organoid models have shown special promise as a bridge between standard two-dimensional in vitro culture models, and in vivo models such as murine xenografts. Traditional two-dimensional cell culture techniques require significant adaptations by cells simply to grow in vitro and also do not recapitulate important features of three-dimensional cellular organization of tissues, including complex cell-cell interactions [10]. Animal models are routinely used in preclinical testing of drugs before human trials. However, such in vivo models generally take many months to establish, are costly to develop and maintain, and are optimal only for low-throughput testing of a small number of drugs. Recent work demonstrated that organoid models of cancer can recapitulate the biology of a variety of tumor types [11,12,13,14]. Additionally, when used in conjunction with drug testing assays, these models can accurately reflect the treatment response observed in patients [15]. By generating organoid models directly from patient tissue, we have the opportunity to establish a biobank of tumors which can reflect the wide range of histologic and molecular tumor types that exist under the umbrella of a single diagnosis such as hepatoblastoma [13]. As an added benefit of this technique, organoid models of non-tumor “normal” tissue can be established concurrently from patients and used as matched controls for a variety of experimental techniques. 

On the basis of previous work establishing the genomic stability and hepatic functionality of liver organoids derived from primary tissue [16,17,18], we successfully developed and characterized a novel organoid model of aggressive hepatoblastoma as a proof of concept for this approach. We demonstrate that hepatoblastoma tumor organoids faithfully recapitulate the mutational profile and tumor architecture observed in vivo in the patient and in a patient-derived xenograft (PDX). We also demonstrate that tumor organoids share similar gene expression patterns with other hepatoblastoma tumors, and that these transcriptomic profiles are unique from both normal tissue, and organoids derived from normal tissue. This hepatoblastoma organoid model also exhibits key features of Wnt/β-catenin signaling that are hallmarks of hepatoblastoma. Furthermore, we show that hepatoblastoma organoids can be used in a screen of potential therapeutics. Further applications of this model, and expansion to include hepatoblastoma organoids from additional patients, have the potential to improve treatment options for hepatoblastoma patients who do not respond to existing treatment modalities.

## 2. Results

### 2.1. Hepatoblastoma Normal and Tumor Organoids are Established from Primary Tumors through Culture at the Time of Surgery

Protocols for the establishment and long-term expansion of human liver organoid cultures have been previously published elsewhere [16,17,18]. Hepatoblastoma tumor tissue and surrounding tissue was collected in the operating room at the time of tumor resection. For six patients, the resection was a primary liver tumor and included the adjacent non-tumor liver. In one patient (HB9), the resection was of a lung metastasis and the surrounding lung tissue. For three patients (HB2, HB3, and HB5), only frozen tissue was collected. This tissue was used only for the evaluation of histology, whole exome sequencing (WES), and transcriptomic profiling with RNA sequence analysis (RNAseq). For four patients (HB6, HB7, HB8, and HB9), both fresh and frozen tissue was collected. 

Frozen tissue was used for histology, WES, and RNAseq, while fresh tissue was used to establish patient-derived xenografts and to grow organoid cultures (see Figure 1 for schematic of the workflow and Methods section for full details). Fresh tissue pieces were mechanically and enzymatically dissociated to a near single-cell suspension and plated on 24-well plates in suspension in three-dimensional matrix domes. The initial media was an isolation media (IM) which contained Wnt3α, R-spondin, Noggin, and inhibitors of TGF-beta and Rho kinase. After the formation of organoids (usually by 5–7 days), cultures were transferred to an expansion media (EM), for at least 5 additional days. Organoid cultures were expanded in EM and passaged as needed (every 5–7 days for normal organoids and every 10–14 days for tumor organoids). All organoid lines had grown for at least 10 passages (at least 60 days). A subset of organoids was transferred to differentiation media (DM) for 14 days to induce hepatocyte differentiation [17]. DM organoids were analyzed for histology and WES, and both EM and DM organoids were used for transcriptome analysis. 

We successfully established six human liver organoid lines from three patients with hepatoblastoma (HB6, HB7, and HB8). After multiple passages, two of the organoids that were derived from tumor tissue (HB6 tumor, and HB7 tumor) did not show evidence of the mutations present in their associated tumor tissue samples. We hypothesize that non-tumor cells in the tissue sample outgrew the tumor cells in culture prior to organoid harvest for analysis, as has been previously reported [14]. Patient-derived xenografts were also established for two lines (HB8 and HB9) through subcutaneous implantation of tumor pieces (see Methods for full details).

### 2.2. Normal and HB Tumor Organoids Show Morphologic and Genomic Similarity to Human Tissue from Which They Are Derived

Hepatoblastoma is divided into several histologic subtypes depending on morphologic features, and tumors often contain a heterogeneous mix of histologic subtypes [19]. The tumors included in our cohort represent a range of histologic subtypes reflecting these histologically observed variations of hepatoblastoma (Table 1).

Additionally, tumors had a variety of risk profiles based on previously published groupings by Sumazin [4], and Cairo [5]. HB3, HB5, HB8, and HB9 harbored high risk features in the prognostic model by Sumazin (see Appendix A and Appendix A), and HB3, HB5, and HB9 also clustered together in analysis of the 16-gene risk signature by Cairo. Hepatoblastoma tumor organoids were generated from a patient with one of these high-risk tumors, HB8. These organoids display similar morphologic features to the tissue from which they were derived, including disorganized tissue structure and the presence of primitive tubules that are characteristic of the embryonal subtype [19] (Figure 2). Tumor organoids also appear notably different in appearance from organoids derived from adjacent non-tumor liver from the same patient. In contrast with the disorganized clusters characteristic of tumor organoids, non-tumor liver organoids form well-organized spheres. At higher magnification, the differences in tumor and normal organoids can be observed at the cellular level. Normal organoids are made up of cells that have relatively uniform size and shape with a nucleus to cytoplasm ratio that is similar to normal hepatocytes. The cells of tumor organoids, similar to the cells from the tumor, are pleiomorphic with non-uniform nuclei and a high nucleus to cytoplasm ratio (NC ratio) that is characteristic of malignancy in general, and hepatoblastoma in particular (Figure 2) [20].

Hepatoblastoma tumors typically show minimal genomic alterations including few somatic mutations and limited copy number alterations. We used whole exome sequencing (WES) to determine the genomic similarity between organoids and the tissues from which they were derived. Our sample of seven hepatoblastoma tumors demonstrated a range of mutations that have been reported in previous genomic descriptions of hepatoblastoma, most notably mutations in CTNNB1, the gene encoding beta-catenin [4,6,21,22]. Tumor organoids harbored the same mutations that were present in the patient tumor that were characteristic of hepatoblastoma. The normal liver organoids did not harbor any of these mutations. Tumor organoids had a large deletion of exon 3 in CTNNB1. Mutations in this region of CTNNB1 were identified in samples from HB3, HB5, HB7 and HB9 in our cohort (Figure 3), consistent with previous reports of such mutations being present in 60–90% of hepatoblastoma tumors [4,6]. Tumor organoids and tumor tissue, but not normal tissue, from HB8 also both had a point mutation in NFE2L2, a gene involved in the regulation of oxidative stress. This mutation has been identified in 5–10% of hepatoblastoma tumors, and has been associated with a more aggressive clinical phenotype [4,22]. 

Copy number variation was also determined by whole exome sequencing by comparing tumor samples to matched normal samples from the same patient (Figure 3). As has been described in other genomic characterizations of hepatoblastoma, there are few consistent copy number variations in the tumors in our cohort. Some tumors displayed previously described alterations including gains in chromosome regions 1q (three of seven tumors), 2 (one of seven, with two of seven showing focal gains in 2q), and 8q (one of seven) [4,23]. Tumor organoids, as well as the PDX derived from the same sample, showed similar copy number variation profiles to the tumor tissue sample from which they were derived. Interestingly, the increase in copy number variations in PDX and organoid samples was greater than in the human tumor tissue samples. This is likely due to the fact that organoid and PDX samples are enriched for tumor cells while tumor tissue samples also contain normal cells and stromal tissue that do not contain the same genomic alterations as tumor cells themselves. Normal organoids did not demonstrate copy number variations that were present in tumor samples. They did show mild gains in small regions of chromosomes 7 and 19 in all three normal organoid lines. The low level of copy number variation profiles of normal organoid samples further indicates that they are similar to the patients’ normal liver from which they were derived.

### 2.3. HB tumor Organoids Recapitulate Beta-Catenin Signaling Patterns Present in Human HB Tumors

Aberrant Wnt/β-catenin signaling is central to the pathogenesis of hepatoblastoma. We performed multiple assays to evaluate elements of the Wnt/β-catenin signaling pathway in organoid samples compared to tissue and found that hepatoblastoma tumor organoids show signaling abnormalities indistinguishable from the tumor tissue. As noted above, at the genomic level, WES demonstrated a mutation resulting in a complete deletion of exon 3 of β-catenin in HB8 organoid and tumor tissue. We probed for the presence of the β-catenin protein product using Western blotting. Tumor tissue, organoids, and PDX, but not normal tissue, all contain a protein band migrating faster than the expected β-catenin band at 90 kDa. This apparent molecular weight corresponds with what is expected from the truncated version of β-catenin without exon 3 (Figure 4C). We further tested this hypothesis by performing Western blotting using an antibody against an epitope on exon 3 of β-catenin. While there was a clear band in normal samples, no protein was detected in the tumor samples. This is consistent with β-catenin missing the exon 3 epitope in tumor samples (Figure 4D). 

The distribution of β-catenin was examined in cells with immunofluorescence. Both normal liver tissue (Figure 4A) and organoids (Figure 4B) from normal tissue display primarily membrane staining. In contrast, tumor tissue and organoids derived from tumor tissue show increased overall immunoreactivity, with much of it cytoplasmic and nuclear, as evidenced by the increased colocalization with the Hoechst stains (Figure 4A,B). β-catenin is active as a transcription factor when it is localized to the nucleus [24,25], suggesting that Wnt signaling is increased in activation in both the tumor and tumor organoid samples. 

We performed Gene Set Enrichment Analysis (GSEA) [26] using a validated set of 50 genes related to Wnt/β-catenin signaling to examine downstream signaling changes in Wnt/β-catenin signaling. This analysis demonstrated that there is a generalized overactivity of the Wnt/β-catenin signaling pathway in hepatoblastoma tumor tissue, tumor organoid, and PDX tumor samples compared to normal tissue (Figure 4E). To further investigate Wnt/β-catenin signaling at the single-gene level we examined patterns in specific transcripts and again found similar expression in tumor tissue and organoid samples (Figure 4F). Interestingly, some Wnt/β-catenin genes also showed expression alterations in normal organoids compared to normal tissues (e.g., β-catenin and AXIN2), while others did not show alterations (e.g., DKK4 and WNT6).

### 2.4. Hepatoblastoma Tumor Organoids Have a Similar Transcriptome to Human Tumor Tissue

To explore gene expression patterns in hepatoblastoma tumors and organoids, we performed RNAseq on all tumor and normal tissue samples (see Methods), as well as tumor and normal organoid, and PDX samples in our cohort (24 unique samples in total). In principal component analysis (PCA) we found that the adjacent non-tumor tissue samples clustered in a tight group with the exception of HB9 (Figure 5A). 

All of the adjacent non-tumor tissue was from the liver, except for HB9, which was collected from the surrounding lung. We suspect that the normal gene expression differences between the lung and liver explain the separation of HB9 from other normal samples. In contrast, tumor samples separated broadly from normal tissue, mostly along principal component 2 (PC2), with tumor organoids and PDX samples clustering the furthest away from normal tissue samples of any of the tumor samples. We hypothesize that the greater separation of the organoids from the normal, relative to the separation of the primary tumor from the normal tissue samples, may be due to higher enrichment of tumor cells in PDX and organoid samples compared to tumor tissue samples. This is similar to what we observed with the greater magnitude of copy number alterations described above.

Interestingly, normal liver organoid samples clustered in a unique group separate from both tumor samples and normal tissue samples. Normal organoid samples were very similar to normal tissue samples on the axis of PC2 (which separates normal and tumor tissue samples), but separated on the axis of PC1. A similar phenomenon was described by Huch and colleagues in their study of hepatocellular carcinoma and cholangiocarcinoma organoids, in which they demonstrated that one principal component dimension distinguished various tumor types, while another separated tissue samples from organoid samples [14]. Interestingly, of all the tumor samples, the tumor organoid samples were shifted furthest on PC1 toward the normal organoid samples. This shift suggests that tumor organoid samples have expression characteristics inherent to both hepatoblastoma tumors (demonstrated by tumor associated PC2 shift away from normal tissue and organoids), and of organoids in general (demonstrated by PC1 shift away from tissue samples and toward other organoids). Our RNAseq cohort included both expansion media (EM) and differentiation media (DM) samples of all organoids. The samples in EM clustered in one group while DM samples clustered in a separate group. The DM group shifted slightly towards the human tissue samples in accordance with their greater similarity to mature hepatocytes.

We performed differential expression analysis between normal tissue and tumor tissue samples and found 3413 genes that were differentially expressed at a false discovery rate (FDR) < 0.05 and |Log2fold change| greater than 1 (Figure 5B). In order to determine whether tumor organoids showed similar transcriptomic alterations to tumor tissue, we performed unsupervised clustering of sample based on this set of 3413 genes differentially expressed in tissue. We found that all tumor samples clustered together. Normal organoids clustered separately from all tumor samples, but also separately from normal tissue samples. This suggests that tumor organoids recapitulate the gene expression patterns of tumor tissue with high fidelity. In contrast, normal organoids predominantly share the gene expression patterns of normal tissue, but do show some variations.

We employed Gene Set Enrichment Analysis (GSEA) to further explore the gene expression patterns that characterized different paired sets of samples [26]. We performed comparisons between tumor samples and normal tissue samples, and between normal EM organoids and normal DM organoids. We queried samples against 50 canonical gene sets that include verified groups of genes associated with well-defined biological states and cellular processes [27]. We found seven gene sets that had enriched expression in tumor samples compared to normal tissue at a nominal *p* value of < 0.05 (Appendix A). The expression of gene sets related to Wnt/β-catenin signaling, MYC target genes, E2F transcription targets, and others were enriched in tumor samples compared to normal samples. In comparing EM and DM samples, DM samples were found to exhibit enrichment of gene sets related to a number of fundamental hepatocyte functions including coagulation, complement, and drug metabolism (Figure 5C, Appendix A). Enrichment of TGF-β signaling in DM samples validates the effect of TGF-β inhibition in EM samples provided by compound A8301 in EM media.

### 2.5. HB Organoids Allow Medium-Throughput Drug Screening and Can Be Used for Individualized Drug Testing

Using normal liver organoids and tumor organoids from the HB8 lines, we performed a proof-of-concept drug screen. This patient had previously received standard-of-care therapies, yet experienced subsequent recurrences and metastases. Thus, we wanted to explore novel compounds that had not previously been used to treat hepatoblastoma. Based on a high-throughput drug screen that we are pursuing against PDX for various liver tumors, we chose 12 candidate compounds from a screen of 5200 compounds [28]. Organoids were treated in dilution series of all compounds for six days and then evaluated using a cell viability assay (Figure 6). Complete results for all compounds are in the Appendix A. The assay was conducted simultaneously with three technical replicates and repeated with later passage organoid populations and organoids transduced with an L10 GFP (again in triplicate) as biological replicates (see Appendix A for full results).

Cisplatin, the cornerstone of standard chemotherapy for hepatoblastoma, was included in the screen in addition to the candidate compounds. The drug had no differential effect on tumor organoids compared to normal organoids, even at high concentrations. A few drugs including the proteasome inhibitor MG-132 and the broad-spectrum kinase inhibitor Staurosporine demonstrated increased killing of normal organoids compared to tumor. Some drugs exhibited greater reduction in tumor organoid populations compared to normal organoids, but only at relatively high concentrations that also resulted in significant death in normal organoids (e.g., a674563 and AZD3463). One drug, JQ1, demonstrated a promising result of increased destruction of tumor organoids compared to normal organoids. This was true at multiple concentrations and suggests there may be an acceptable therapeutic index for JQ1 in hepatoblastoma. 

## 3. Discussion

There is a critical need for novel models to study hepatoblastoma [7]. In addition to the small number of cell lines for the disease, there is significant inconsistency in the literature characterizing some of these models. For example, the HepG2 cell line, which is widely used in hepatoblastoma research, was cultured from a patient significantly older than usual hepatoblastoma patients (15 years old) and is alternately referred to as a hepatocellular carcinoma cell line or a hepatoblastoma cell line in different publications [29,30,31,32]. In order to develop targeted therapies for hepatoblastoma, disease models should faithfully reflect the diversity of histological and molecular subtypes to specifically target the more aggressive forms of the disease.

Recently, three-dimensional organoid culture models have transformed in vitro studies of human biology and disease. Organoids have allowed scientists to recapitulate normal and disease physiology at a multicellular level that facilitates both single-cell analysis and the understanding of cellular interactions across heterogenous cellular populations analogous to the heterogenous makeup of organs in living animals. Thus, organoid models have the potential to fulfill their promise as a missing link between the two-dimensional cell culture models that lack the richness of cell interactions in animals, and in vivo models that prohibit high throughput approaches. Cancer organoids have now been shown to faithfully recapitulate key elements of disease including sequential mutation [33], treatment response [14,15], and host immune response to the tumor [34]. Organoid biobanks have been established as a proof-of-concept for a number of tumor types including colon cancer [13] and breast cancer [35]. These biobanks have the potential to be used to direct the treatment of individual patients in a precision medicine approach [36], or can be used to guide treatment for broader groups based on molecular or mutational profiles [13]. The objective of using phenotypically diverse laboratory models to guide novel treatments for aggressive pediatric liver cancer is a goal shared by a number of authors in the field. Nicolle and colleagues demonstrated the feasibility of using a bank of patient-derived xenografts to predict clinical outcomes and select appropriate therapies in pediatric liver cancer [37]. Additionally, a recently published study by Eloranta and colleagues used a panel of cell-line-derived aggressive hepatoblastoma spheroids for drug testing in a similar fashion [38].

Our study presents a novel organoid model of hepatoblastoma, the most common pediatric liver tumor. Moreover, this model represents a particularly aggressive subtype of the disease that is most often associated with poor outcomes. This aggressive subtype is under-represented in existing hepatoblastoma models [7], and is the subject of special focus for novel therapeutic techniques [39]. We have demonstrated that hepatoblastoma tumor organoids show morphologic similarity to the tumors from which they were derived, and retain genomic and transcriptomic alterations associated with the tumors. Wnt/β-catenin signaling aberrations, which are fundamental to the pathophysiology of hepatoblastoma, were also retained between tumor tissue and organoids. While the yield of tumor organoid lines was relatively low in our study (one of four fresh samples collected), this may be improved with culture techniques optimized to select for tumor organoids similar to those presented by Broutier and colleagues [14].

In addition to our organoid lines, we also developed two murine patient-derived xenografts from patient samples in our cohort. The results from tumor histology, exome sequencing, gene expression, and β-catenin protein expression experiments suggest similar biology between our organoid and PDX models of hepatoblastoma. Organoids have the advantage that they can be generated within weeks of resection, unlike a PDX, which could take up to a year before being validated. Thus, the organoid can provide drug-response data in a time frame that could directly inform patient care. The in vivo murine PDX models are also costlier and more labor intensive to maintain, further strengthening the advantages of organoid models as a surrogate. However, PDX models have the unique advantage of modeling the tumor in a living animal and thus maintain a key role in validating results from experiments first carried out in organoids.

Until now, organoid models have not been widely used to study pediatric cancer. This is likely due to the fact that pediatric cancers are comparatively rare, comprising less than 1% of all cancer cases diagnosed in the United States. This is despite the fact that cancer is the second leading cause of death among individuals younger than 20 years of age [40]. Organoids have unique promise in pediatric cancer, however, specifically due to the rarity of pediatric tumors and the dearth of validated in vitro models. In the case of hepatoblastoma, a biobank of hepatoblastoma tumors with unique molecular features has the potential to guide treatment for patients with extremely rare tumor subtypes that would likely not be captured in standard drug screening using cell lines and xenografts. Our initial proof-of-concept drug screen identified JQ1 as a compound with promising activity against our aggressive hepatoblastoma tumor organoid model. JQ1 is an inhibitor of the bromodomain and extraterminal domain (BET) family of proteins, and has shown promise in pre-clinical trials in a wide variety of tumors, including another common pediatric solid tumor, neuroblastoma [41]. The mechanism of action of BET inhibitors is related to epigenetic regulation of the cell cycle with specific effects on the MYC oncogene. NMYC amplification is a fundamental element of the aggressive form of neuroblastoma [42] and there is evidence that MYC activation plays a role in the aggressive phenotype in hepatoblastoma as well [5,43]. Our tumor organoids showed significant MYC signaling enrichment in GSEA. Other BET inhibitor molecules have been altered to improve pharmacologic characteristics for delivery in human patients and are being used in phase I and II clinicals trials in both solid tumors and hematologic malignancies [44]. BET inhibitors have not yet been applied to hepatoblastoma but have been recently shown to have some effect in improving the response to immunotherapy in mouse models of hepatocellular carcinoma [45]. These compounds may represent new opportunities for therapy in aggressive forms of hepatoblastoma.

## 4. Materials and Methods 

Supplementary methods are found in Appendix B.

### 4.1. Human Specimens

With Institutional Review Board approval (Rockefeller IRB# SSI-0797, SSI-0798 and Memorial Sloan Kettering Cancer Center IRB Protocol #13-010), we obtained frozen OCT embedded tumor tissue and adjacent normal tissue from 3 patients (HB2, HB3, and HB5) and fresh tumor tissue and adjacent normal tissure from 4 patients (HB6, HB7, HB8, and HB9) (Table 1). In all patients, diagnosis of hepatoblastoma was confirmed by an independent pathologist at the time of patient tumor resection. Fresh tissue samples were separated and processed for histology, RNA and DNA isolation, implantation into mice to produce xenografts, or dissociated and processed for organoid culture.

### 4.2. Isolation and Culture of Human Normal Liver and Tumor Organoids

Normal liver and tumor organoids were isolated and cultured using minor modifications of a protocol previously published by Huch et al. [16,17,18]. See supplemental methods for additional details. 

### 4.3. Histology and Immunofluorescence

Standard methods for histology and immunofluorescence staining and imaging were used. See supplemental methods for additional details. 

### 4.4. RNA Isolation and Sequencing

RNA was extracted from organoids in culture or from OCT embedded frozen hepatoblastoma tumor and adjacent normal liver tissue samples using the miRNeasy Mini Kit (Qiagen, Germantown, MD, USA). Organoids were mechanically disrupted and incubated in organoid harvesting solution (Trevigen, Gaithersburg, MD, USA) at 4 °C for 1 hour prior to extracting RNA. RNA sequencing libraries were generated using the TruSeq Stranded Total RNA Sample Prep Kit with Ribo-Zero ribosomal RNA depletion (Illumina, San Diego, CA, USA). Libraries were sequenced with 2 × 150 bp paired-end reads on an Illumina NextSeq system. 

### 4.5. RNA Sequencing Analysis

Quality control of the sequencing files was performed using FastQC and MultiQC [46], followed by adapter trimming using BBDuk (included in BBMap v37.47). Sequences were then mapped to the annotated Human Genome GRCh38.92 using STAR v2.6.1. The read counts per gene were analyzed using R v3.5.1 and Rstudio v1.1.456. Differential expression analysis was performed using DESeq2 v1.18.1 [47] studying the sample type (normal tissue vs. tumor tissue). We selected those genes with *p*-value (FDR) < 0.05 and log_2_|FoldChange| > 1. Box and scatter plots of the normalized gene counts (log_2_ transformed) were generated for each gene of interest.

### 4.6. Exome Capture and Sequencing

DNA from frozen normal and tumor tissue and DNA from normal and tumor organoids were extracted using the DNeasy Blood and Tissue Kit (Qiagen, Germantown, MD, USA), according to the manufacturer’s protocol. After PicoGreen quantification and quality control by Agilent BioAnalyzer, 197–250 ng of DNA was used to prepare libraries using the KAPA Hyper Prep Kit (Kapa Biosystems KK8504) with 8 cycles of PCR. After sample barcoding, 114–500 ng of library was captured by hybridization using the xGen Exome Research Panel v1.0 (IDT) according to the manufacturer’s protocol. PCR amplification of the post-capture libraries was carried out for 8 cycles. Samples were run on a HiSeq 4000 in a 100 bp/100 bp paired end run, using the HiSeq 3000/4000 SBS Kit (Illumina). Normal and tumor samples were covered to an average of 138 X and 172 X, respectively. Reads were aligned to the reference human genome (hg38). Resulting BAM files were processed for quality control and mutations were detected using MuTect 2 [48]. The results were visualized using Interactive Genomics Viewer (IGV v2.4.9, Broad Institute, Cambridge, MA, USA).

### 4.7. Drug Screening

Drug screening was performed using a protocol previously published [49]. Briefly, organoids were cultured as described above and were prepared for drug screening as follows. Organoids were mechanically disrupted and counted as per the above-mentioned protocol. Organoids were diluted in liver organoid expansion media with 2% BME-2 to about 15,000–20,000 organoids per mL. A layer of matrix (10 µL of 7.5 mg/mL BME-2) was dispensed onto the bottom of each well of 384-well plates and incubated at 37 °C until solidified. Thirty microliters of the organoid solution was dispensed into each well of the 384-well plate on top of the solidified matrix. The plated organoids were incubated overnight at 37 °C, 5% CO2. The next day, 10 µL of the compound (12 drugs: cisplatin, JQ1, tryphostin 9, a674563, AZD3463, MG-132, RITA, OTX015, omaveloxolone, staurosporine, ivermectin, and CCT137690) (Selleckchem, Houston, TX, USA) was added to each well at differing concentrations (0.005, 0.01, 0.04, 0.1, 0.37. 1.1, 3.3, and 10 µM). After 6 days of incubation, cell viability was quantified using CellTiter-Glo 3D Cell Viability Assay (Promega, Madison, WI, USA). The drug screen was performed in triplicate on biological duplicates of differing passage number. The cell viability ratios of drugged to untreated cells were plotted by average and standard deviation across the differing concentrations of drug. 

### 4.8. Mouse Xenograft Studies

Nod-Scid-Gamma (NSG) mice were purchased from Jackson Laboratories (Bar Harbor, NOD.Cg-Prkdcscid Il2rgtm1Wjl/SzJ (005557)) and bred at The Rockefeller University Center for Comparative Biology SPF immune-core. With IACUC approval (#14681-H), mice were anesthetized using isoflurane and given buprenorphine for analgesia. Fresh tumor pieces obtained from human subjects were implanted subcutaneously on the flank bilaterally. Female and male mice were used, and all mice were 6–8 weeks of age at time of tumor implantation. All interventions were performed during the light cycle. Mice were fed LabDiet #5053—amoxicillin diet (LabDiet, St. Louis, MO, USA) and kept in microvent individually ventilated cages (IVC) (Allentown, Allentown, NJ, USA) with corn-cob bedding and paper enrichment. Mice were followed for general health and for tumor growth, and tumors were passaged to new host mice based on deteriorating health or tumor size based on the standard criteria for maintaining tumors in rodents.

### 4.9. Immunoblotting

Standard immunoblotting methods were used. See supplemental methods for details.

## 5. Conclusions

In conclusion, our study demonstrates the feasibility of using patient-derived organoids as a laboratory model for aggressive hepatoblastoma. We have developed such a model and validated its similarity to hepatoblastoma tumor tissue using histology, exome sequencing, RNA sequencing, and protein expression assays including immunoblotting and immunofluorescence. This model has the additional advantage of the ability to use non-tumor organoids derived from the same patient as a matched control. We have demonstrated that these organoids can be used successfully for medium-throughput drug screening and, in doing so, identified a candidate compound, the BET inhibitor JQ1, for further study in therapeutic applications for hepatoblastoma. In addition to establishing an organoid model of hepatoblastoma, this study demonstrates the promise of organoids to model other relatively uncommon pediatric solid tumors that have few validated laboratory models.

## Figures and Tables

**Figure 1 cancers-12-02668-f001:**
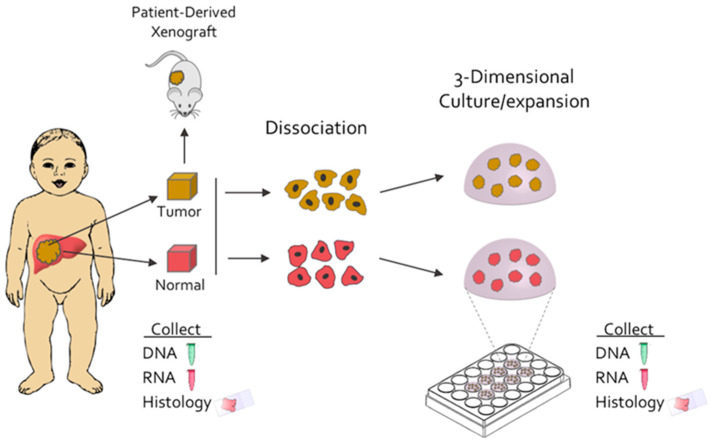
Fresh hepatoblastoma tumor tissue and surrounding normal tissue is collected from patients at the time of surgery. The tissue is dissociated to single-cell suspension and cultured in a 3-dimensional matrix to create tumor and normal liver organoids. Pieces of fresh tumor are also implanted subcutaneously into mice to generate PDXs. Collection of DNA and RNA, and histologic examination is performed at multiple points in the experimental workflow.

**Figure 2 cancers-12-02668-f002:**
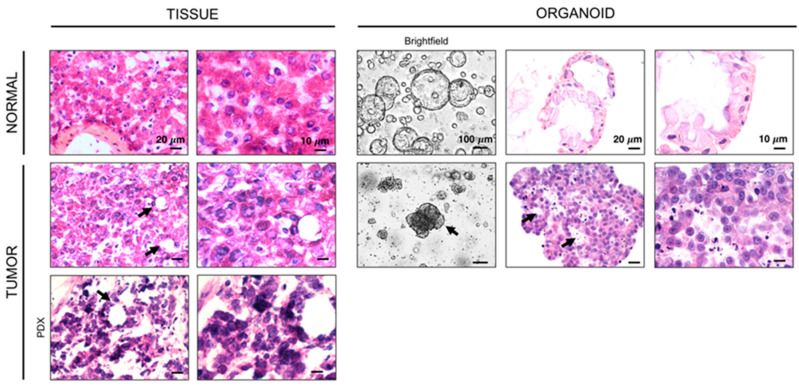
Representative hematoxylin and eosin staining of normal and tumor tissue and organoids from a patient with hepatoblastoma (HB8). Healthy liver and primary tumor tissue sections were obtained from frozen sections of tissue blocks that were used for whole exome sequencing (WES) and RNAseq. All images were obtained with a 60× objective except brightfield images which were obtained with a 10 × objective. Normal tissue and organoid samples show relatively uniform cell size and shape, while hepatoblastoma tumor tissue and tumor organoids have a similar disorganized appearance and include immature tubular structures (dark arrows).

**Figure 3 cancers-12-02668-f003:**
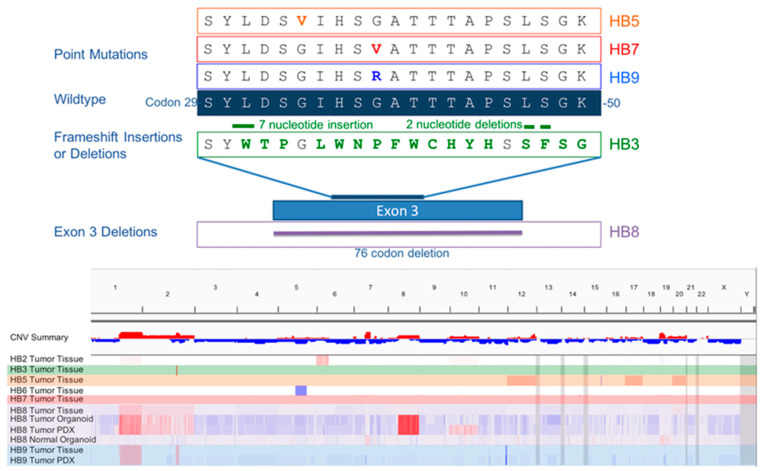
Mutations in exon 3 of CTNNB1 were found in 4 out of 7 tumor samples. Three tumors had a single point mutation. HB8 tumor had a complete deletion of exon 3, while another (HB3) had multiple frameshift insertion/deletions. A copy number variation plot for all tissue and organoid samples demonstrates that tumors reflect known hepatoblastoma copy number variations. These variations are similarly reflected in tumor organoid and PDX samples but not in normal organoid samples.

**Figure 4 cancers-12-02668-f004:**
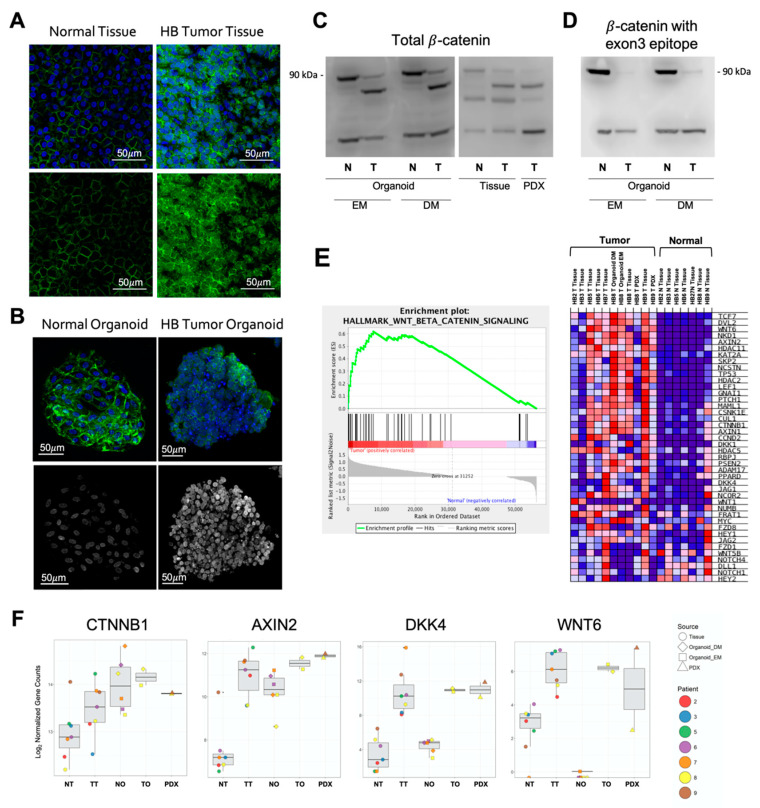
(**A**,**B**) Immunofluorescence images demonstrate that hepatoblastoma tumor tissue and tumor organoids have greater cytoplasmic and nuclear beta-catenin staining than normal tissue and normal organoids. Green = β-catenin, blue = Hoechst, gray = co-localization. Scale bars = 50 µm. (**C**) Immunoblot of total β-catenin in normal organoids and tissue shows full length protein at 90 kDa and truncated β-catenin in HB8 tumor organoids, tissue, and PDX associated with large deletion of exon 3. (**D**) Immunoblot of β-catenin using an antibody with an epitope on exon 3 demonstrates the loss of staining in HB8 tumor organoids that harbor an exon 3 deletion. Detail information can be found at Appendix A. (**E**) Gene Set Enrichment Analysis (GSEA) of the Wnt/β-catenin pathway in normal tissue compared to all tumor samples demonstrates a significant increase in the transcription of Wnt/β-catenin signaling genes in tumor samples. (**F**) Wnt/β-catenin gene expression patterns at the single gene level compared between normal tissue (NT), tumor tissue (TT), normal organoids (NO), tumor organoids (TO), and patient derived xenografts (PDX) demonstrate similar expression between tumor tissue and tumor organoid samples.

**Figure 5 cancers-12-02668-f005:**
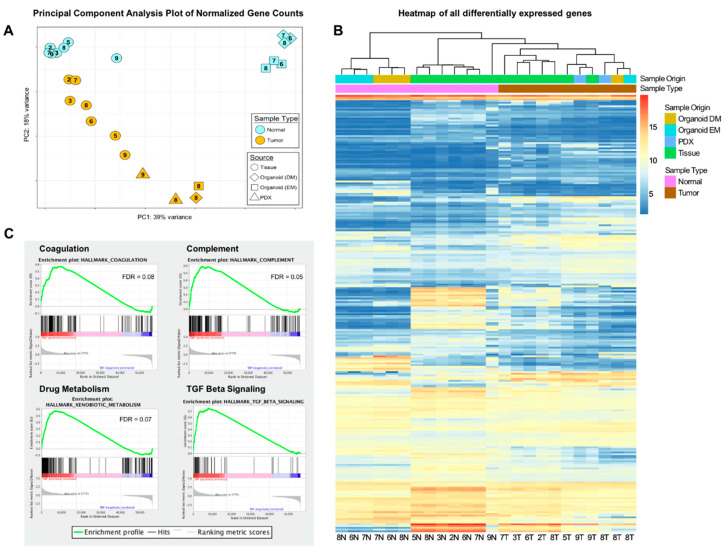
(**A**) Principal component analysis (PCA) plot of dimensions 1 and 2 (PC1 vs PC2) for RNAseq data (normalized counts per gene). The plot demonstrates similarity between normal tissue samples and a separation of normal organoids along PC1 and tumor samples along PC2. (**B**) Heatmap showing differentially expressed genes (*n* = 3413) between hepatoblastoma tumor tissue samples and normal tissue samples. Individual genes are represented by rows, and samples are represented by columns. Columns are organized by unsupervised clustering. All normal samples cluster together as do all tumor samples, however normal organoids cluster as a distinct group from normal tissues. (**C**) Gene Set Enrichment Analysis comparing organoids in EM and DM. DM organoids show increased activity in a variety of normal hepatocyte functions and enriched TGF-β signaling. Organoid expansion media contains a TGF-β inhibitor, while DM does not.

**Figure 6 cancers-12-02668-f006:**
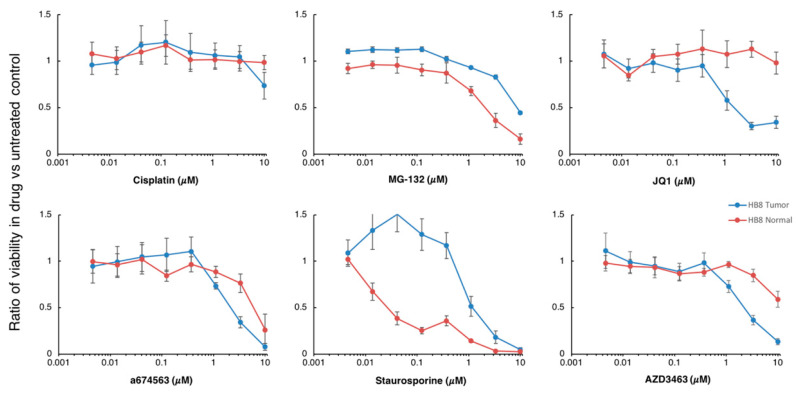
Organoid drug screen results for 6 selected compounds. Normal and tumor organoids were tested simultaneously for multiple concentrations between 1 nM and 10 μM. Plots shown are the mean of three biologic replicates each performed in triplicate at each dilution. Error bars represent standard error of all replicates. Cisplatin which is commonly used to treat hepatoblastoma showed no differential effect on tumor organoids compared to normal. JQ1 demonstrated increased killing of tumor organoids compared to normal organoids and may be candidate for novel therapy in aggressive hepatoblastoma.

**Table 1 cancers-12-02668-t001:** Frozen tumor and surrounding normal tissue were collected from seven patients. The table shows the histologic subtypes of tumors included in the sample cohort, and the grey boxes indicate the material collected for each patient sample.

Sample	Histologic Subtype	Frozen Tissue	Fresh Tissue	Normal Organoid	Tumor Organoid	PDX
HB2	Fetal					
HB3	Mixed epithelial and mesenchymal					
HB5	Fetal and embryonal					
HB6	Fetal					
HB7	Mixed epithelial and mesenchymal					
HB8	Predominantly fetal with minor embryonal					
HB9	Mixed embryonal and fetal

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
