# Peer review of "A Human Organoid Model of Aggressive Hepatoblastoma for Disease Modeling and Drug Testing"

_cancers, 2020, doi:10.3390/cancers12092668_

Round 1

Reviewer 1 Report

The authors demonstrated the feasibility of using patient-derived organoids as a laboratory model for aggressive hepatoblastoma.

The manuscript is well written, but I have some comments.

1. Specimens were collected from 7 cases, and tumor organoid was established in 1 case, but the reason why it is inefficient has not been fully discussed. Please comment on the problem with this method.

2. In Figure 3, there seems to be a wide range of mutations in CTNNBI for each case. Is the HB8 tumor organoid established this time universal as a hepatatoblastoma tumor organoid?

3. PCA of hepatoblastoma varies greatly in Figure 5A. Can you think that HB8 is a typical hepatoblastoma?

4. The right side of Figure 5B is cut and cannot be seen. Please correct.

5. It seems that the drugs examined in Figure 6 are not the drugs commonly used for hepatoblastoma, but what about the sensitivity to vincristine and 5-FU?

Author Response

Reviewer 1:
1. Specimens were collected from 7 cases, and tumor organoid was established in 1 case, but the reason why it is inefficient has not been fully discussed. Please comment on the problem with this method.

Frozen specimens were collected for all 7 cases, but fresh tissue was only collected from 4 cases (HB 6, 7, 8, and 9). Of these 4 cases, normal liver organoids were established for HB 6, 7, and 8. HB9 was a lung metastasis and thus we did not attempt to create normal lung organoids. We were successful in culturing organoids from HB6 and HB7 tumor samples, but when these organoids were analyzed after multiple passages using exome sequencing, they did not harbor the mutations present in the parent tumors, and did not cluster with the other tumor samples in RNAseq analysis. Since this culture technique was originally optimized to culture normal liver organoids, we hypothesize that normal cells present in the tumor tissue sample also formed organoids that became dominant after multiple passages. This phenomenon was also reported by Broutier et al. The culture method has been adapted to culture other liver tumors (Broutier 2018) which
may lead to a greater success rate with hepatoblastoma organoids as well.

To clarify this issue in the text, an additional column for “Fresh tissue” has been added to table 1 (line 143). Additionally, the phrasing describing this issue was edited to improve clarity as follows:
“After multiple passages, two of the organoids that were derived from tumor tissue (HB6 tumor, and HB7 tumor) did not show evidence of mutations present in their associated tumor tissue samples. We hypothesize that non-tumor cells in the tissue sample outgrew the tumor cells in culture prior to organoid harvest for analysis, as has been previously reported [14].” (Lines 115 – 119)

An additional sentence was added to the discussion as well:
“While the yield of tumor organoid lines was relatively low in our study (1 of 4 fresh samples collected) this may be improved with culture techniques optimized to select for tumor organoids similar to those presented by Broutier and colleagues [14].” (Lines 355-368)

2. In Figure 3, there seems to be a wide range of mutations in CTNNBI for each case. Is the HB8 tumor organoid established this time universal as a hepatoblastoma tumor organoid?

While no single tumor line can represent a universal tumor organoid, we do suggest that the HB8 tumor organoid represents specific features unique to some aggressive hepatoblastoma (e.g. the patient had treatment resistant disease, and the tumor contains and NFE2L2 mutation). The greater promise of this model is the ability to create different organoid lines that will represent the range of genomic and clinical variations of hepatoblastoma.
Text in the discussion section (line 339) has been altered to emphasize that this model represents an aggressive subtype rather than being a universal model of aggressive hepatoblastoma.

3. PCA of hepatoblastoma varies greatly in Figure 5A. Can you think that HB8 is a typical hepatoblastoma?

Heterogeneity in PCA results among tumor samples is to be expected due to a variety of factors including tumor subtype, varying treatment effect, and presence of stromal cells in tissue samples. Rather than presenting HB8 as a typical hepatoblastoma, the message that our PCA plot is intended to communicate is that normal and tumor samples represent different clusters, and the tumor and normal organoid samples derived from the same patient cluster in different groups.

We have added the word “broadly” to emphasize the spread among the tumor samples in line 250.

4. The right side of Figure 5B is cut and cannot be seen. Please correct.

This has been corrected. Full size TIFF images of all figures have also been uploaded.

5. It seems that the drugs examined in Figure 6 are not the drugs commonly used for hepatoblastoma, but what about the sensitivity to vincristine and 5-FU?

The goal of our drug screen was to identify novel compounds that may impact aggressive hepatoblastoma tumors that don’t respond to conventional treatment rather than evaluate the effectiveness of current treatments. We did however include cisplatin (the first line therapy for Hepatoblastoma in both COG and SIOPEL protocols) in our screen and demonstrated no sensitivity among the HB8 organoids.
We have edited the text as follows to emphasize the exploratory nature of the drug
screen:
“This patient had previously received standard-of-care therapies, yet experienced
subsequent recurrences and metastases. Thus, we wanted to explore novel compounds that had not previously been used to treat hepatoblastoma.” (lines 294-296) and

“Cisplatin, the cornerstone of standard chemotherapy for hepatoblastoma was included in the screen in addition to the candidate compounds.” (lines 312-313)

Reviewer 2 Report

This manuscript from Sanford Simon’s group describes generation and characterization of organoids from patients with aggressive hepatoblastoma (HBL). Therapeutic approaches for chemo-resistant, aggressive hepatoblastoma are not available due to lack of good biological systems for the drug screen. In this manuscript, the authors have established six human organoid cell lines from patients with HBL and two patient derived xenografts from freshly resected patients. They also generated organoids from normal livers. The authors found that the tumor organoids showed morphological and genomic similarity (beta-catenin and NRF2 mutations) to human tissues from which they were derived. The additional studies revealed that HBL organoids have similar transcriptome and b-catenin/WNT signaling as the original tumors have. The authors also demonstrated that the established HBL organoids are excellent tools for screening of drugs. In fact, they identified JQ1 as a promising drug which shows high potentials to destruct HBL organoids, but not organoids from healthy livers. This work is highly innovative and highly significant for development of therapy for patients with aggressive HBL. The manuscript will be of great interest for the readers of Cancers.

I have two minor comments/suggestions.

First,

In addition to organoids, the authors established two PDXs from HB8 and HB9. The data from PDXs could be better integrated by inclusion of a paragraph in the section “Discussion”, which will summarize common features of generated PDXs and organoids.

Second,

The discussion can be improved by citation of two recent papers. One paper by Eloranta et al (Frontiers in Oncology 2020) describes the use of HBL derived spheroids in the search for drugs. The second paper by Liu et al (Cancer Science 2020) describes potentials of the JQ1 to improve immunotherapy in HCC. The discussion of the results of these two reports and the results of the current manuscript would improve the manuscript.

Author Response

Reviewer 2:
1. In addition to organoids, the authors established two PDXs from HB8 and HB9. The data from PDXs could be better integrated by inclusion of a paragraph in the section “Discussion”, which will summarize common features of generated PDXs and organoids.

We have added the following paragraph to the discussion section (lines 359-368):
“In addition to our organoid lines, we also developed two murine patient derived
xenografts from patient samples in our cohort. Results from tumor histology, exome sequencing, gene expression, and β-catenin protein expression experiments suggest similar biology between our organoid and PDX models of hepatoblastoma. Organoids have the advantage that they can be generated within weeks of resection, unlike a PDX which could take up to a year before being validated. Thus, the organoid can provide drug-response data in a time frame that could directly inform patient care. The in vivo murine PDX models are also costlier and more labor intensive to maintain, further strengthening the advantages of organoid models as a surrogate. However, PDX models have the unique advantage of modeling the tumor in a living animal and thus maintain a key role in validating results from experiments first carried out in organoids.”

2. The discussion can be improved by citation of two recent papers. One paper by Eloranta et al (Frontiers in Oncology 2020) describes the use of HBL derived spheroids in the search for drugs. The second paper by Liu et al (Cancer Science 2020) describes potentials of the JQ1 to improve immunotherapy in HCC. The discussion of the results of these two reports and the results of the current manuscript would improve the manuscript.

We have added references to these interesting articles in the discussion section as
follows:
“A recently published study by Eloranta and colleagues used a panel of cell-line-derived aggressive hepatoblastoma spheroids for drug testing in a similar fashion [36].” (Lines 344-346) “BET inhibitors have not yet been applied to hepatoblastoma but have been recently shown to have some effect in improving the response to immunotherapy in mouse models of hepatocellular carcinoma [43].” (Lines 404-405)
